# Bioactive Constituents of *Verbena officinalis* Alleviate Inflammation and Enhance Killing Efficiency of Natural Killer Cells

**DOI:** 10.3390/ijms24087144

**Published:** 2023-04-12

**Authors:** Xiangdong Dai, Xiangda Zhou, Rui Shao, Renping Zhao, Archana K. Yanamandra, Zhimei Xing, Mingyu Ding, Junhong Wang, Tao Liu, Qi Zheng, Peng Zhang, Han Zhang, Yi Wang, Bin Qu, Yu Wang

**Affiliations:** 1State Key Laboratory of Component-Based Chinese Medicine, Tianjin University of Traditional Chinese Medicine, Tianjin 301617, China; 2Department of Biophysics, Center for Integrative Physiology and Molecular Medicine (CIPMM), School of Medicine, Saarland University, 66421 Homburg, Germany; 3Leibniz Institute for New Materials, 66123 Saarbrücken, Germany; 4Institute for Immunology, School of Medicine, Tsinghua University, Beijing 100084, China; 5School of Integrative Medicine, Tianjin University of Traditional Chinese Medicine, Tianjin 301617, China; 6Pharmaceutical Informatics Institute, College of Pharmaceutical Sciences, Zhejiang University, Hangzhou 310058, China

**Keywords:** *Verbena officinalis*, natural killer cells, killing efficiency

## Abstract

Natural killer (NK) cells play key roles in eliminating pathogen-infected cells. *Verbena officinalis* (*V. officinalis*) has been used as a medical plant in traditional and modern medicine for its anti-tumor and anti-inflammatory activities, but its effects on immune responses remain largely elusive. This study aimed to investigate the potential of *V. officinalis* extract (VO extract) to regulate inflammation and NK cell functions. We examined the effects of VO extract on lung injury in a mouse model of influenza virus infection. We also investigated the impact of five bioactive components of VO extract on NK killing functions using primary human NK cells. Our results showed that oral administration of VO extract reduced lung injury, promoted the maturation and activation of NK cells in the lung, and decreased the levels of inflammatory cytokines (IL-6, TNF-α and IL-1β) in the serum. Among five bioactive components of VO extract, Verbenalin significantly enhanced NK killing efficiency in vitro, as determined by real-time killing assays based on plate-reader or high-content live-cell imaging in 3D using primary human NK cells. Further investigation showed that treatment of Verbenalin accelerated the killing process by reducing the contact time of NK cells with their target cells without affecting NK cell proliferation, expression of cytotoxic proteins, or lytic granule degranulation. Together, our findings suggest that VO extract has a satisfactory anti-inflammatory effect against viral infection in vivo, and regulates the activation, maturation, and killing functions of NK cells. Verbenalin from *V. officinalis* enhances NK killing efficiency, suggesting its potential as a promising therapeutic to fight viral infection.

## 1. Introduction

*Verbena officinalis* (*V. officinalis*), also known as common vervain, is a medicinal herb that is widely distributed in the temperate climate zone across the globe [1]. In China, *V. officinalis* is widely distributed in the southern part of the Yellow River and has been used for centuries as traditional Chinese medicine to treat rheumatism, bronchitis, depression, insomnia, anxiety, liver, and gallbladder diseases [2,3]. Moreover, it has a long-standing record of use in food and cosmetics, which validates its safety [3]. The bioactive constituents of V. officinalis mainly include flavonoids, terpenoids, phenolic acids, phenylpropanoids, and iridoids [2,3,4,5]. Recent reports have shown various activities of *V. officinalis*, including anti-oxidation, anti-bacteria, anti-fungi, and anti-inflammation properties [1,6,7,8].

Inflammation is an immune response that can be triggered by a range of factors, including virus, bacteria, and transformed cells. During the inflammatory process, the permeability of blood vessels is enhanced, which facilitates the recruitment of immune cells to the site of inflammation. The recruited immune cells release cytokines that further activate and recruit other effector immune cells. Inflammatory responses are essential for fighting pathogens; however, uncontrolled inflammatory responses can have severe consequences, such as organ dysfunction, particularly in the lungs, and the potentially life-threatening cytokine storm syndrome. Innate immune cells are the primary initiators of inflammatory responses.

In the innate immune system, natural killer (NK) cells are specialized immune killer cells, which play key roles in eliminating tumorigenic and pathogen-infected cells. In response to virus infection, NK cells are recruited to the lung and contribute to the immune response to fight pathogens. Several studies have highlighted the pivotal role of NK cells in controlling influenza virus infection. Defects in NK cell activity or depletion of NK cells result in delayed viral clearance, increased morbidity, and mortality [9,10]. However, there are also examples in which NK cells exacerbate morbidity and pathology during lethal dose influenza virus infection in mice [11,12]. These suggest that overactivation of NK cells may lead to undesirable effects. In addition, NK cells are important in bridging the innate and adaptive immune responses to virus infection [13].

In this work, we investigated the anti-inflammatory effect of VO extract in vivo using virus-infected mice with low (0.5 g/kg) and high (1 g/kg) doses. We found that both doses significantly reduced the release of inflammatory cytokines (TNFα, IL-6, and IL-1β), induced by virus infection. Notably, the low dose provided better protection of the lung tissue and induced a higher level of NK activation. Further analysis of bioactive constituents from *V. officinalis* revealed that Verbenalin substantially enhanced the killing efficiency of NK cells.

## 2. Results

### 2.1. VO Extract Attenuated the Acute Lung Damage Induced by Virus Infection

Infections caused by viruses, such as influenza or SARS-CoV (severe acute respiratory syndrome coronavirus)-1/2, can result in lung damage, leading to severe breathing problems or even respiratory failure [14]. Acute lung damage caused by virus infection may also result in persistent lung abnormalities, including pulmonary fibrosis, which can lead to long-term respiratory impairments in patients who have recovered from the infection [15]. This type of lung damage is, to a large extent, owing to the massive inflammation initiated by an overactivated immune system. The well-established anti-inflammatory activity of *V. officinalis* prompted us to examine its effect on infection-induced lung damage. To this end, we infected C57BL/6J mice with the influenza virus A/PR8/34 (H1N1) and orally administered VO extract upon viral infection once per day for 3 days (Figure 1A). We chose the low dose (0.5 g/kg) and the high dose (1 g/kg) based on the doses in previous studies in mice [2,16], without any detectable toxicity to tissues, including heart, liver, and kidneys (Appendix A). Body weight was monitored daily, and we observed loss of weight in the virus-infected group, which was significantly alleviated by VO extract administration for both the low and high doses (Figure 1B). Using H&E staining of lung sections, we examined changes in alveolar morphology and immune cell infiltration on day 3 post-viral infection. We observed massive immune cell infiltration, thickening of alveolar walls, and disrupted lung parenchyma in virus-infected mice (Figure 1C, Virus group vs. Control group). However, these infection-induced symptoms in the lung were considerably reduced by VO extract administration for both the low and high doses (Figure 1D, Low/High group vs. Virus group). Interestingly, the low dose appeared to achieve a better attenuation of symptoms than the high dose (Figure 1D, Low group vs. High group). Moreover, the administration of VO extract reduced the release of inflammatory cytokines (TNF-α, IL-1β and IL-6) triggered by viral infection to a level comparable to that of the Control group (Figure 1E–G). No significance was identified between the high and low doses (Figure 1E–G). Taken together, our findings indicate that VO extract is a potent agent for reducing infection-induced acute lung damage and inflammatory response.

### 2.2. VO Extract Promoted Maturation and Activation of NK Cells in the Lungs in Response to Viral Infection

NK cells are key players in the elimination of pathogen-infected cells. To evaluate the effect of VO extract on NK functions, we examined the frequency and activation of NK cells isolated from lung tissues on day 3 post-infection (Figure 2A). We found no significant alteration in the frequency of lung-residing NK cells following VO extract administration (Figure 2B). To further evaluate NK cell activation, we used surface markers CD11b and CD69, as their expression is indicative of the effector functions and cytotoxicity of murine NK cells [17,18]. We found that viral infection substantially enhanced the frequency of the CD11b^+^ and CD69^+^ NK subsets (Figure 2C,D). Interestingly, this tendency was further elevated by the administration of the low dose of VO extract, but not by the high dose (Figure 2C,D). These results indicated that VO extract at a low dose specifically promoted the activation of NK cells during viral infection.

### 2.3. Identification of Chemical Composition from VO Extract by UPLC-Q-TOF-MS

To identify the active ingredients in VO extract, we employed ultra-high-performance liquid chromatography-quadrupole time-of-flight mass spectrometry (UPLC-Q-TOF-MS). A representative base peak chromatogram (BPC) of VO extract in positive and negative ion modes was shown in Figure 3. We successfully identified thirteen ingredients, including 3,4-dihydroverbenalin, Verbeofflin I, Hastatoside, Verbenalin, Quercetin, Acteoside, Luteolin, Isorhamnetin, Luteolin 7-O-β-gentiobioside, Isoacteoside, Leucosceptoside A, Apigenin, and Kaempferol (Appendix A). At the same time, we determined the relative content of key bioactive components in VO extract (Appendix A).

### 2.4. Bioactive Components of V. officinalis Enhanced NK Killing Efficiency

To assess the effect of bioactive compounds from *V. officinalis* on NK cell functions, we cultured primary human NK cells with specific compounds (10 µM and 30 µM) in the presence of IL-2 for three days. We first analyzed the killing kinetics of NK cells using a plate-reader-based real-time killing assay [19]. We found that Acteoside, Apigenin, and Kaempferol slightly reduced NK cells’ killing efficiency, whereas Verbenalin and Hastatoside enhanced it (Figure 4A). Notably, Verbenalin exhibited the highest potency in elevating NK cells’ killing efficiency (Figure 4A). In addition, the presence of three negative bioactive constituents (Acteoside, Apigenin, and Kaempferol) did not counteract the effect of Verbenalin on NK cell killing (Appendix A). To further confirm the effect of Verbenalin on NK cell killing, we performed a 3D killing assay. Target cells expressing FRET-based apoptosis reporter pCasper were embedded in a collagen matrix and NK cells were added from the top. Target cells were yellow when alive and turned green when undergoing apoptosis [20]. Our results showed that Verbenalin-treated NK cells exhibited significantly faster killing kinetics, compared to their counterparts treated with the vehicle. Thus, we concluded that Verbenalin was capable of increasing NK cells’ killing efficiency under physiologically relevant conditions, and among the bioactive constituents of *V. officinalis*, it has the most significant impact on NK cells’ killing function.

### 2.5. Verbenalin Accelerated NK Killing Processes

Next, we sought out the underlying mechanisms regulating the increase in NK killing efficiency by Verbenalin. We examined proliferation, expression of cytotoxic proteins (perforin and granzyme B), and degranulation of lytic granules. We found that Verbenalin did not significantly impact those processes (Figure 5A–C). We then analyzed the time required for killing by visualizing the killing events every 70 s for 12 h using high-content imaging (Figure 6A, Movie S1). The time required for killing is defined as the duration from the initiation of NK/target contact to the target cell apoptosis. The quantification showed that for the NK cells treated with 30 µM of Verbenalin, the time required for NK cells to kill was considerably reduced (Figure 6B). Concomitantly, the number of target cells killed per NK cell was almost doubled in the Verbenalin-treated NK cells relative to their vehicle-treated counterpart (Figure 6C). It has been reported that a portion of NK cells can serve as serious killers, which are able to kill several target cells in a row [21,22]. We thus also analyzed the frequency of serial killers (NK cells that killed more than one target cell), single killers (NK cells that killed only one target cell), and non-killers (NK cells that did not kill any target cells). We found that Verbenalin treatment substantially increased the portion of serial killers while decreasing the portion of non-killers (Figure 6D). For the NK cells treated with 10 µM of Verbenalin, the time required for NK cells to kill was also reduced (Appendix A), with no significant change in the number of target cells killed per NK cell (Appendix A). The frequency of serial killers was enhanced and that of non-killers was decreased (Appendix A). In summary, our results suggested that Verbenalin potentiated NK cell activation upon target recognition and shortened the time required to initiate target cell destruction, leading to an increase in killing efficiency of NK cells.

To explore underlying mechanisms, we investigated potential interaction of Verbenalin with NK inhibitory receptors, especially NKG2A and KIR2DL1, using an in silico molecular docking analysis. Our results showed that Verbenalin could bind to both NKG2A and KIR2DL1 (Appendix A). This finding suggested that Verbenalin might enhance NK cell killing by reducing inhibitory receptor-mediated signaling, thus facilitating the rapid activation of NK cells for efficient elimination of target cells.

## 3. Discussion

Uncontrolled immune responses induced by infection are often associated with life-threatening consequences, such as respiratory failure due to lung damage and cytokine storm syndrome. In this process, exacerbated inflammatory responses initiated by innate immunity play an essential role. Thus, early interventions that minimize undesired inflammation without compromising immune responses to fight pathogens are of great significance to achieve optimal clinical outcomes. In this work, we demonstrated that the extract of *V. officinalis*, a medical herb with a long history of utilization in traditional Chinese medicine and alternative medicine in Western countries, significantly reduced viral infection-induced acute lung damage as well as release of proinflammatory cytokines. At the same time, administration of a low dose of VO extract considerably enhanced NK activation in response to viral infection. In addition, we identified that Verbenalin, a biologically active constituent of *V. officinalis*, substantially elevated NK cell-mediated cytotoxicity by shortening the time required for killing and, consequently, enhancing the frequency of serial killers. These findings suggest that *V. officinalis* and Verbenalin are promising therapeutic agents for early intervention to protect lung function, avoid cytokine storm, and facilitate clearance of virus-infected cells.

The beneficial effect of VO extract on lung injury may arise from multiple mechanisms. Previous studies have shown that treatment with *V. officinalis* inhibits the replication of respiratory syncytial virus [23], and that treatment of active constituents of *V. officinalis* increases phagocytotic activity of neutrophils in vitro [23]. In our study, we administered VO extract orally, and its concentration in the lungs might have reached levels that affected viral replication to some extent. We postulated that the enhanced phagocytic activity of neutrophils in the VO extract-treated group might have contributed to the efficient removal of viral particles from the lungs. Although our study used a mouse model of influenza virus infection, we speculated that the effect of VO extract on acute lung injury induced by viral infection would be applicable to other respiratory viral infections that result in severe respiratory complications. This hypothesis is bolstered by a recent study that reported that a newly developed formula, Xuanfei Baidu, composed of thirteen medical herbs including *V. officinalis* has shown very positive clinical outcomes in treating patients with SARS-CoV-2 infection [23,24]. In addition, CD8+ cytotoxic T lymphocytes (CTLs) play critical roles in the adaptive immune defense against viral infection. However, we would like to emphasize that the effect we have observed was on day 3, which is earlier than the peak of the primary murine CTL response upon influenza virus infection, typically observed on day 5 [25]. It is worth noting that NK cells can mount a cytotoxic response without prior priming, allowing them to respond to a viral infection immediately. Therefore, it is unlikely that CTLs are the primary targets of VO extract in attenuating the acute lung damage induced by viral infections.

Release of proinflammatory cytokines triggered by viral infection recruits immune cells to inflammation sites. In this study, we found that VO extract administration led to reduced levels of proinflammatory cytokines, including TNF-α, IL-1β, and IL-6, in serum. TNF-α is primarily released by M1-type macrophages and T cells [26], and is mainly regulated by the NF-κB pathway [27]. IL-1β is commonly released by monocytes, macrophages, and mast cells; however, non-immune cells, such as epithelial cells, endothelial cells, fibroblasts, and neuronal and glial cells, can also synthesize and release IL-1β during cell injury or inflammation [28]. IL-1β is regulated by the NF-κB, c-Jun N-terminal kinase (JNK), and p38 MAPK pathways [29]. IL-6 can be released by myocardial and immune cells [30], and is primarily triggered and regulated by signaling pathways such as NF-κB and MAPK [31]. It is reported that total glucosides of *V. officinalis* attenuate chronic nonbacterial prostatitis in rat models by reducing the release of IL-2, IL-1β, and TNF-α in the prostate [32]. Additionally, Verbenalin, a bioactive constituent of *V. officinalis*, can effectively reduce airway inflammation in asthmatic rats by inhibiting the activity of the NF-κB/MAPK signaling pathway [33]. These pathways and molecules are possible targets for *V. officinalis* to regulate the release of proinflammatory cytokines.

In this work, we observed that Verbenalin-treated NK cells were able to destroy target cells more quickly than their vehicle-treated counterparts. To successfully execute their killing function, NK cells need to identify their target cells using surface receptors, followed by the formation of a tight junction, termed the immunological synapse (IS), between the NK cell and the target cells. At the IS, lytic granules (LGs) containing cytotoxic proteins, including pore-forming protein perforin and serine protease granzymes, are enriched and released to induce target cell destruction [34,35]. Thus, the time required for killing is determined by several steps: engagement of surface-activating/inhibitory receptors, formation of the IS, enrichment and release of LGs, and uptake of cytotoxic proteins by target cells.

Both activating and inhibitory receptors are expressed on NK cells to control their activation [36]. Engagement of activating receptors, such as NKp46, NKp30, NKp44, and NKG2D, triggers signaling pathways for NK activation [37]. Inhibitory receptors bind to major histocompatibility complex (MHC) Class I molecules, which are expressed on healthy self-cells. Loss or down-regulation of MHC-I molecules leads to activation of NK cells to initiate killing processes [38]. Formation of the IS between NK cells and target cells largely depends on the interaction between LFA-1 and ICAM-1 [39]. Enrichment of LGs at the IS is regulated by reorganization of the cytoskeleton, especially reorientation of MTOC towards the contact site [40]. LG release requires proper docking at the plasma membrane and vesicle fusion with the plasma membrane, which are tightly regulated by SNARE and related proteins [41,42]. Uptake of cytotoxic proteins by target cells requires Ca^2+^-dependent endocytosis [43]. Enhancement in any of the above-mentioned steps could accelerate killing processes, such as sensitizing activating receptors, up-regulating effector molecules downstream of activating receptors, promoting LG accumulation at the IS, reducing the dwell time for docking, or enhancing the efficiency of LG release. Our results suggested that Verbenalin bound to NK inhibitory receptors NKG2A and KIR2DL1, indicating that enhancement of NK cell killing by Verbenalin could be a result of its ability to reduce inhibitory signaling. Engagement of surface receptors is important to initiate and regulate the following steps: IS formation, LG enrichment and release, etc. Further investigation is needed to characterize how each step is affected by Verbenalin to accelerate killing.

## 4. Materials and Methods

### 4.1. Preparation of VO Extract

*Verbena officinalis* was obtained from Anhui Zehua China Pharmaceutical Slices Co., Ltd. The whole plant (4.5 kg) was extracted with 4.5 L of 70% ethanol for 2 h by refluxing extraction repeated three times. The combined extract was filtrated with ceramic membrane and concentrated using vacuum evaporation apparatus at a temperature not exceeding 45 °C. The resulting extract was then lyophilized to obtain the VO extract powder.

### 4.2. UPLC-Q-TOF-MS Analysis

Quantitative analysis was performed using an Agilent 1290 UHPLC system (Agilent Technologies Inc., Palo Alto, CA, USA) coupled to an Agilent 6520 Q-TOF instrument with electrospray ionization (ESI) source. Chromatographic separation was achieved using an ACQUITY UPLC^®^ BEH C18 (2.1 × 150 mm, 1.7 μm; Waters, Milford, MA, USA). The mobile phase consisted of 0.1% aqueous formic acid (A) and methanol (B). The elution condition involved holding the starting mobile phase at 95% A and 5% B and applying a gradient of 5% A and 95% B for 35 min. The flow rate was set at 0.3 mL/min, and the injection volume for all the sample was 2 µL. Experiments were performed in positive and negative ESI mode with the following parameters: ESI temperature, 100 °C; collision energy, 10 V; collision pressure, 135 V; fragmentor voltage, 135 V; nebulizer gas, 40.0 psi; dry gas, 11.0 L/min at a temperature of 350 °C; scan range, *m*/*z* 100–1700.

### 4.3. Quantification of the Active Compounds from VO Extract

The prepared samples were injected into a Waters ACQUITY UPLC system (Waters, Milford, MA, USA) with a photodiode array (PDA) detector. The chromatographic separation was performed with a BEH C18 column (2.1 mm × 150 mm, 1.7 μm, Waters), operated at 35 °C, and the sample injection volume was set at 2 μL. The flow rate was kept constant at 0.3 mL/min and UV measurements were obtained at 254 nm. The mobile phases were methanol (solvent A) and water containing 0.1% formic acid (solvent B) with gradient elution using the following program: 0–35 min, 5–95% A.

A 10 mg/mL sample solution was formed from weighed VO extract powder and added methanol, swirled for 30 s, ultrasonic dissolved for 3 min, and centrifuged at 13,000 rpm for 10 min. The supernatant was filtered by 0.22 μm microporous filter membrane and then sampled to inject for UPLC and LC-MS analysis.

The mixed standard solution was composed of methanol solution with 0.1 mg each of hastatoside, verbenalin, and acteoside, and 0.01 mg each of apigenin and kaempferol per mL.
C = Pr × Cx

C: the concentration of the compounds in the sample; Pr: peak area ratio of compounds in sample and in mixed standard; Cx: the concentration of compounds in a mixed standard.

### 4.4. Mice and Virus

Female C57BL/6 mice (6–10 weeks old, weighing 20 to 25 g) were purchased from Beijing Vital River Laboratory Animal Technology Co., Ltd. (Beijing, China), and housed in standard microisolator cages in a centralized animal care facility. Animal care and experimental procedures were performed in accordance with experimental animal guidelines. Mice were given ad libitum access to food and water and subjected to a 12 h light/dark cycle. All mice were adapted to the environment for seven days before the experiments. Virus (H1N1) was stored at −80 °C.

### 4.5. Virus Infection

Mice were anesthetized with isoflurane and intranasally inoculated with 100 PFU H1N1 in 40 μL PBS. The number of mice in each group ranged from 6 to 8. The Vehicle group received an intranasal challenge with 40 μL PBS. All the animal experiments were approved, and efforts were made to minimize suffering and to reduce the number of animals used.

### 4.6. Administration of VO Extract

To administer the VO extract, VL (0.5 mg/kg) and VH (1 mg/kg) doses were given orally via gavage once daily for three consecutive days. The VO extract was freshly prepared each day and stored at 4 °C until administration. Distilled water was orally administered to mice in the Vehicle group simultaneously. On the third day, the mice were sacrificed, and their blood and lung tissues were collected for further analysis. During these three days, mice were monitored and changes in body weight were recorded.

### 4.7. Flow Cytometry Assay

To perform the flow cytometry analysis, nonspecific receptors were blocked with anti-CD16/CD32, and then surface markers (CD45, NK1.1, CD11b, and CD69) were stained. Cells were fixed and then incubated with specific surface-binding antibodies for 30 min at 4 °C. Samples were analyzed using BD FACScalibur and FlowJo software. Cells were gated according to forward scatter and side scatter, and NK cells were identified by the CD45^+^NK1.1^+^ population.NK1.1^+^CD11b^+^ were used to determine maturation of NK cells and NK1.1^+^CD69^+^ were used to determine activation of NK cells.

### 4.8. Hematoxylin–Eosin (HE) Staining

To assess pathological changes, the mice were sacrificed on the third day, and their lung, heart, liver, and kidney tissues were collected, fixed in 10% buffered formalin, and embedded in paraffin. Each tissue was cut into 4 μm sections and stained with hematoxylin and eosin. Lung injury was evaluated according to a quantitative scoring system that assesses infiltration of immune cells, thickening of alveolar walls, and disruption of lung parenchyma [44,45,46]. The scoring for lung damage was performed by three researchers independently according to standard protocols.

### 4.9. Detection of IL-6, TNF-α, and IL-1β Levels in Serum

To obtain the serum, blood samples were centrifuged at 3000 rpm and 4 °C for 15 min. The concentrations of IL-6, TNF-α, and IL-1β in serum were measured using corresponding ELISA kits (Sino Best Biological Technology Co., Ltd., Shanghai, China) according to the manufacturer’s instructions with a microplate reader (Tecan Trading AG, männedorf, Switzerland).

### 4.10. NK Celsl Preparation and Cell Culture

Primary human NK cells were isolated from peripheral blood mononuclear cells (PBMCs) of healthy donors using the Human NK Cell Isolation Kit (Miltenyi Biotec, Bergisch Gladbach, Germany). The isolated NK cells were cultured in AIM V medium (Thermo Fischer Scientific, Waltham, MA, US) with 10% FCS and 100 U/mL of recombinant human IL-2 (Miltenyi Biotec, Bergisch Gladbach, Germany). K562 and K562-pCasper cells were cultured in RPMI-1640 medium (Thermo Fischer Scientific, Waltham, MA, US) with 10% FCS. For K562-pCasper cells, 1.25 mg/mL G418 was added. All cells were kept at 37 °C with 5% CO_2_.

### 4.11. Real-Time Killing Assay

The real-time killing assay was conducted as previously reported [19]. Briefly, target cells (K562 cells) were loaded with Calcein-AM (500 nM, Thermo Fisher Scientific, Waltham, MA, US) and settled into a 96-well plate (2.5 × 10^4^ target cells per well). NK cells were subsequently added with an effector-to-target (E:T) ratio of 2.5:1, if not otherwise specified. Fluorescence intensity was determined using the bottom-reading mode at 37 °C every 10 min for 4 h with a GENios Pro microplate reader (TECAN). The target lysis percentage was calculated using the formula:Target lysis (t) % = 100 × (F_live_(t) − F_exp_(t))/(F_live_(t) − F_lysed_(t)). (F: fluorescence intensity)

### 4.12. 3D Killing Assay and Live Cell Imaging

Target cells (K562-pCasper cells) were embedded into 2 mg/mL of pre-chilled neutralized bovine type I collagen solution (Advanced Biomatrix) in a 96-well plate. The collagen was solidified at 37 °C with 5% CO_2_ for 40 min, after which NK cells were added to the top of the collagen as effector cells. The cells were visualized using ImageXpress Micro XLS Widefield High-Content Analysis System (Molecular Devices) at 37 °C with 5% CO_2_. For the 3D killing assay, as described previously [47], killing events were visualized every 20 min for 36 h, and live target cell numbers were normalized to hour 0 based on area. For live cell imaging to determine time required for killing and the average kills per NK cell, the cells were visualized every 70 s for 14 h and tracked manually. ImageJ software was used to process and analyze the images.

### 4.13. Proliferation Assay

To examine proliferation, freshly isolated primary human NK cells were labelled with CFSE (1 μM, Thermo Fischer Scientific, Waltham, MA, US) and then stimulated with recombinant human IL-2 in the presence of Verbenalin at indicated concentrations for 3 days. Fluorescence was determined with a FACSVerse™ flow cytometer (BD Biosciences, San Jose, CA, US) and analyzed with FlowJo v10 (FLOWJO, Ashland, OR, US).

### 4.14. Determination of Cytotoxic Protein Expression

To test perforin and granzyme B expression, NK cells were fixed in pre-chilled 4% paraformaldehyde. Permeabilization was carried out using 0.1% saponin in PBS containing 0.5% BSA and 5% FCS. FACSVerse™ flow cytometer (BD Biosciences, San Jose, CA, US) was used to acquire data. FlowJo v10 (FLOWJO, Ashland, OR, US) was used for analysis.

### 4.15. CD107a Degranulation Assay

For the degranulation assay, K562 cells were co-cultured with vehicle-treated or Verbenalin-treated NK cells in the presence of Brilliant Violet 421™ anti-human CD107a (LAMP1) antibody (Biolegend, San Diego, CA, US) and GolgiStop^TM^ (BD Biosciences). The incubation was carried out at 37 °C with 5% CO_2_ for 4 h. The cells were then stained with PerCP anti-human CD16 antibody (Biolegend) and APC mouse anti-human CD56 antibody (BD Biosciences) to define NK cells. Data were obtained with a FACSVerse™ flow cytometer (BD Biosciences) and analyzed with FlowJo v10 (FLOWJO, LLC).

### 4.16. Molecular Docking

To investigate the interactions between small molecules and receptor proteins, the CDOCKER module in Discovery Studio software was used for molecular docking. The three-dimensional structures of verbenalin molecules were downloaded from TCMSP (https://tcmsp-e.com/, accessed on 13 March 2023), and then we performed hydrogenation through the Prepare Ligand module and optimized the energy with the CHARMM force field. The three-dimensional structure of the target protein was downloaded from the PDB database (https://www.rcsb.org/, accessed on 13 March 2023). Then we ran the Prepare Protein module to optimize the protein structure: deleting redundant protein conformations, deleting water molecules, and completing incomplete residues, hydrogenation, and distribution of related charges. The prepared target proteins and small molecules were introduced into Discovery Studio and docked using the CDOCKER module. The semi-flexible docking method and simulated annealing algorithm were used to find the optimal conformation of ligand and receptor. According to the level of CDOCKER Interaction Energy, we evaluated the degree of docking: the lower score function value indicating the stronger affinity between compound and its receptor.

### 4.17. Statistical Analysis

Data were analyzed using SPSS version 19.0 and GraphPad Prism software 5.0 and presented as mean ± standard deviation (SD). Significant differences among the multiple group comparisons were performed using one-way analysis of variance (ANOVA), and the ANOVA comparisons were analyzed through Tukey’s honest significant difference test. Data with a partial distribution were examined using the nonparametric Kruskal–Wallis test with Dunn’s multiple comparison as a post-test. A *p* value less than 0.05 indicated a significant difference.

## 5. Conclusions

The present study provided evidence that VO extract had the ability to ameliorate lung injury induced by viral infection and enhance the maturation and activation of NK cells in the lungs. Moreover, our in vitro study with primary human NK cells showed that Verbenalin significantly reduced the required contact time for killing, thereby enhancing the total number of killing events per NK cell. The findings of this study established a direct link between Verbenalin, a bioactive constituent of VO extract, and the killing efficiency of NK cells, suggesting its promising potential for therapeutic applications in combating viral infections and, potentially, cancer.

## Figures and Tables

**Figure 1 ijms-24-07144-f001:**
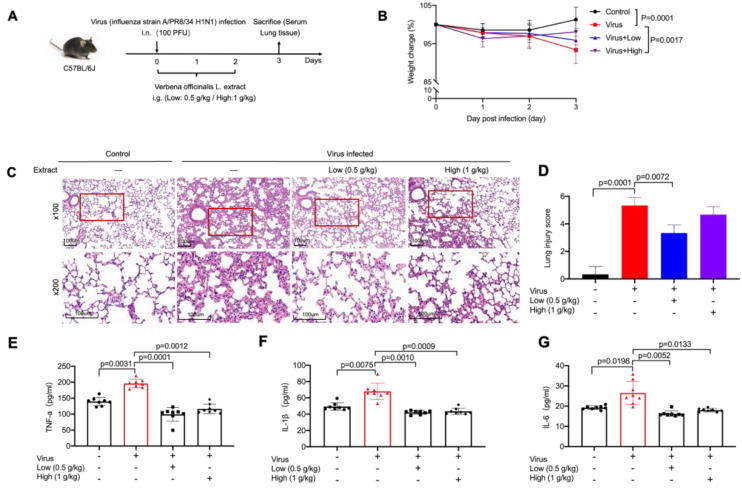
Analysis of the effects of VO extract on infection-induced acute lung damage and inflammation in vivo. (**A**) C57BL/6J mice were intranasally challenged with the influenza virus A/PR8/34 (H1N1) (100 PFU) on day 0 and then orally administered a single dose of VO extract (low dose/Low: 0.5 g/kg; high dose/High: 1 g/kg) every day for 3 days. Mice were sacrificed on day 3. (**B**) Loss of body weight caused by viral infection was ameliorated by VO extract. The body weight of mice was measured daily for three days. *n* = 8 for each group. (**C**) Administration of VO extract alleviated virus-induced inflammation in the lung. Histological analysis of lung tissues was carried out on day 3. Two magnifications are shown: 100× (scale bars: 100 µm) and 200× (scale bars: 100 µm). One representative sample from each group is shown (*n* = 3). (**D**) A total of 50 alveoli were counted on each slide at ×200 magnification (*n* = 3). (**E**–**G**) Administration of VO extract abolished viral infection-triggered release of proinflammatory cytokines. Blood samples were taken on day 3. Cytokine concentration was determined using ELISA. Circles, triangles, squares and diamonds represent the number of mice in each group. Statistical analysis was performed using SPSS version 19.0 and GraphPad Prism software 5.0. Results were presented as mean ± SD.

**Figure 2 ijms-24-07144-f002:**
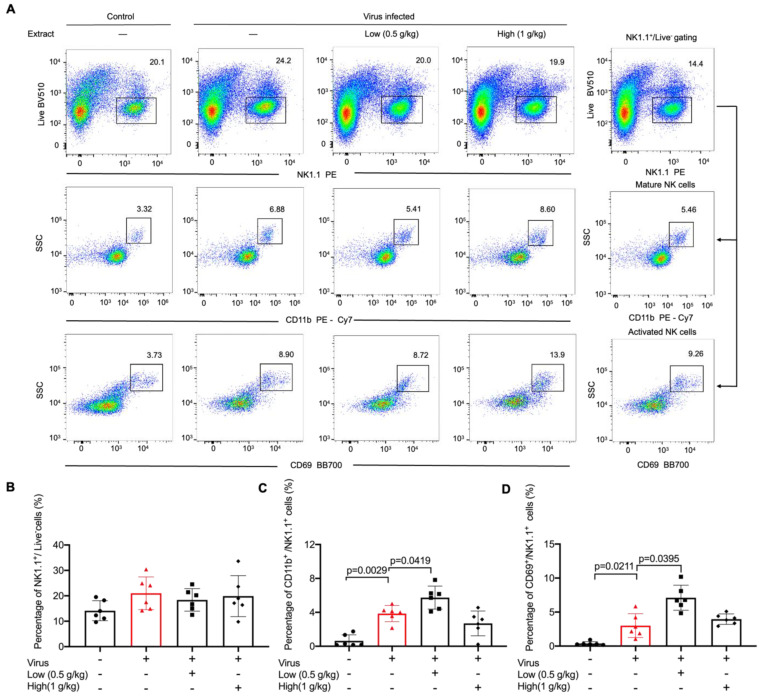
Low dose of VO extract enhances NK activation in response to viral infection. C57BL/6J mice were intranasally challenged with influenza virus A/PR8/34 (H1N1) (100 PFU) on day 0, followed by daily oral administration of a single dose of VO extract (Low: 0.5 g/kg; High: 1 g/kg) for 3 days. On day 3, lung samples were collected, homogenized, and filtered with a 70μm cell filter. (**A**) The cell suspension was stained with control antibodies (PE−conjugated anti−NK1.1 antibody, PE−Cy7−conjugated anti−CD11b antibody, BB700−conjugated anti−CD69 antibody) and analyzed using flow cytometry. (**B**) The CD45^+^NK1.1^+^ population was gated for NK cells. (**C**) NK1.1^+^CD11b^+^ were used to determine maturation of NK cells. (**D**) NK1.1^+^CD69^+^ were used to determine activation of NK cells. Circles, triangles, squares and diamonds represent the number of mice in each group. Statistical analysis was performed using SPSS version 19.0 and GraphPad Prism software 5.0. Results are presented as mean ± SD (*n* = 6).

**Figure 3 ijms-24-07144-f003:**
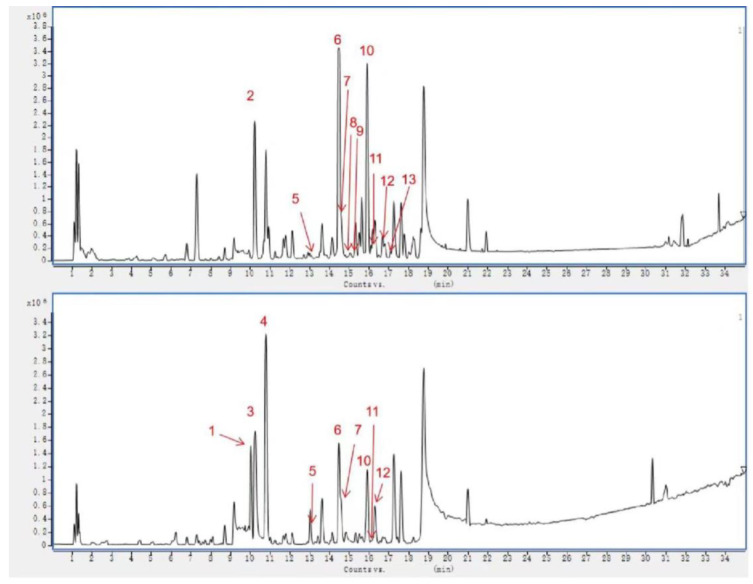
Chemical base peak intensity (BPI) chromatogram of key compounds in VO extract characterized in positive and negative ion modes. UPLC-Q-TOF/MS was employed. The identified compounds were numbered as follows: 3,4-dihydroverbenalin (1); Verbeofflin I (2); Hastatoside (3); Verbenalin (4); Quercetin (5); Acteoside (6); Luteolin (7); Isorhamnetin (8); Luteolin 7-*O*-*β*-gentiobioside (9); Isoacteoside (10); Leucosceptoside A (11); Apigenin (12); and Kaempferol (13). See Appendix A for additional details.

**Figure 4 ijms-24-07144-f004:**
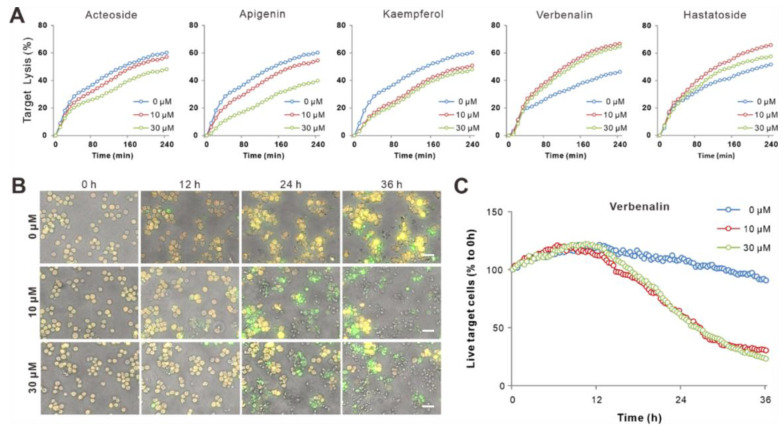
Effect of bioactive constituents of VO extract on NK cells’ killing efficiency. (**A**) Primary human NK cells were cultured with different compounds of *V. officinalis* (10 µM and 30 µM) for three days in the presence of IL-2, and their killing kinetics were determined using a plate-reader-based real-time killing assay. (**B**,**C**) Verbenalin accelerates NK killing kinetics in 3D. K562-pCasper target cells were embedded in collagen matrices, and NK cells were added from the top. The killing events were visualized at 37 °C every 20 min for 36 h. Yellow indicates live target cells. Turning green indicates that they were undergoing apoptosis. Fully lysed target cells lost fluorescence signals. Selected time points are shown in (**B**), and the change in live target cells is shown in (**C**). Scale bars are 40 μm. Results from one representative donor out of four is shown.

**Figure 5 ijms-24-07144-f005:**
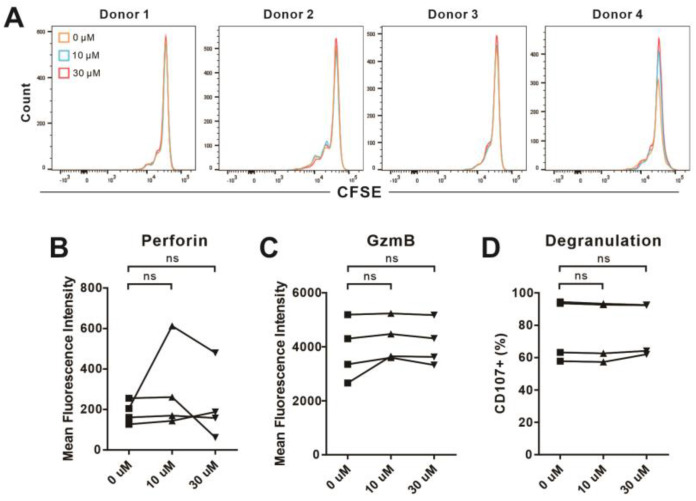
Verbenalin did not affect the proliferation and lytic granule pathway of NK cells. Primary human NK cells were stimulated with IL−2 in the presence of Verbenalin at the indicated concentrations for 3 days prior to experiments. (**A**) Proliferation of NK cells. Freshly isolated NK cells were stained with CFSE and then cultured as described above. (**B**,**C**) Expression of cytotoxic proteins. On day 3, NK cells were fixed, permeabilized, and stained with BV510 anti−human perforin antibody and PerCP/Cyanine5.5 anti-human granzyme B antibody. (**D**) Release of lytic granules was determined by CD107a degranulation assay. Results are shown as a percentage of CD107a^+^ NK cells. Fluorescence was analyzed using flow cytometry and FlowJo. Results were from four donors. Statistical analysis was performed using paired Student’s *t*-test. ns: not significant.

**Figure 6 ijms-24-07144-f006:**
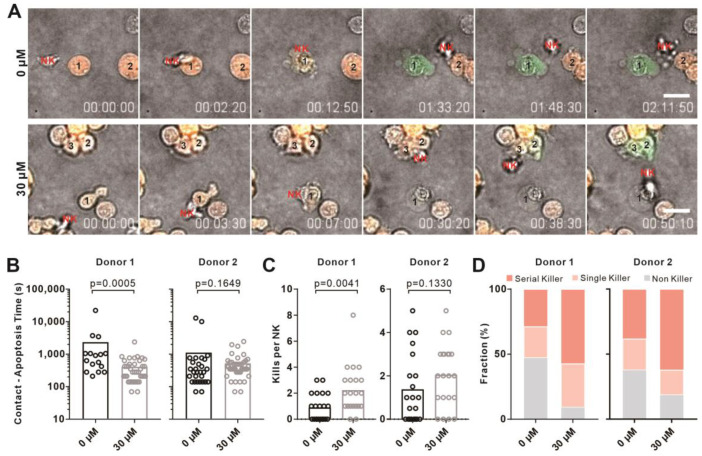
Verbenalin shortened the time required for NK cell killing and increased killing events per NK cell. Primary human NK cells were stimulated with IL−2 in the presence of Verbenalin (30 µM) at the indicated concentrations for 3 days prior to experiments. K562-pCasper target cells were embedded in collagen and NK cells were added from the top. Killing events were visualized at 37 °C every 70 s for 14 h. (**A**) NK cells made multiple contacts with target cells. One representative NK cell from each condition was shown. NK cells (marked in red) were not fluorescently labeled. The corresponding target cells in contact were numbered. Scale bars were 20 μm. (**B**) Verbenalin shortened the time required for NK cell killing. The time from NK/target contact to target cell apoptosis was quantified. (**C**) Verbenalin treatment enhanced the number of target cells killed per NK cell. Both apoptosis and necrosis were considered as killing events. (**D**) The fraction of serial killers was elevated by Verbenalin treatment. The fraction of serial killers (NK cells that killed more than one target cell), single killers (NK cells that killed only one target cell), and non−killers (NK cells that did not kill any target cells) for each donor was analyzed. Results were from two donors. A total of 21 NK cells were randomly chosen from each condition. Statistical analysis was performed using a Mann−Whitney test.

## Data Availability

The datasets used and/or analyzed during the current study are available from the corresponding author upon reasonable request.

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
