# Peer review of "Bioactive Constituents of *Verbena officinalis* Alleviate Inflammation and Enhance Killing Efficiency of Natural Killer Cells"

_ijms, 2023, doi:10.3390/ijms24087144_

Round 1

Reviewer 1 Report

The manuscript "Bioactive constituents of Verbena officinalis alleviate inflammation and enhance killing efficiency of NK cells" by Dai et al. presents interesting findings on the mechanism of action of this widely used medical plant. The study is original and relevant to the scientific and medical communities. However, it  can be improved to be suitable for publication in the International Journal of Molecular Sciences.

Major comments:

1. The authors observed a pronounced anti-inflammatory effect of VO-extract in a relevant model of viral respiratory infection, even when used at low dose. To unveil the underlying mechanism, they decided to analyzed the NK immune response in the lungs. However, they overlooked the contribution of a very prominent player in viral infection, namely CD8+ cytotoxic effector T cells. Understanding the effect of VO-extract on these antiviral T cells in the context of flu infection would greatly benefit the impact and relevance of this study.

2. The authors made an exhaustive analysis of VO bioactive constituents. They found that 3 of them (Acteoside, Apigenin, Kaempferol) slightly reduced NK killing efficiency as opposed to Verbenalin and Hastatoside that clearly enhanced NK function. Which is the relative content (%) of each of these components in VO-extract? Based on that information, could any of the 3 bioactive constituents mentioned above (Acteoside, Apigenin, Kaempferol) counteract the effect of Verbenalin and Hastatoside? Running additional NK killing assays with a mixture of Verbenalin plus any of the 3 negative bioactive constituents might help to understand this issue.

3. The observations related to the acceleration of NK killing by Verbenalin are very interesting but intriguing. This reviewer wonders about the molecular mechanism responsible for the acceleration of the process, that greatly enhances the serial killer activity of NK cells. In the discussion, authors provide some speculative hypotheses that could account for this behavior. In order to gain better insight into the mechanism of action of Verbenalin, this reviewer thinks that authors could try to tackle any of these plausible hypotheses.

Minor comments:

- Some of the citations are listed as "Error! Reference source not found" throughout the text. Please, correct.

Author Response

Dear Reviewers,

We are submitting the revised version of our manuscript (ijms-2215600), titled ‘Bioactive Constituents of Verbena Officinalis Alleviate Inflammation and Enhance Killing Efficiency of Natural Killer Cells’ for consideration of publication in the International Journal of Molecular Sciences.

We are grateful to the reviewers for your valuable and constructive feedback. To address the reviewers' concerns, we have: 1) assessed the potential toxicity of the doses of VO-extract in mice (new Supplementary Fig. 1); 2) determined the relative contents of the five Verbena officinalis bioactive constituents (new Supplementary Fig. 2) and conductedadditional killing assays with a mixture of Verbenalin plus the three negative bioactive constituents (Acteoside, Apigenin, and Kaempferol) (new Supplementary Fig. 3); 3) added a prediction of the interaction between Verbenalin and NK receptor NKG2A and KIR2DL1 using an in silico molecular docking analysis (new Supplementary Fig. 4); 4) re-written the Discussion section to improve the cohesiveness, clarity, and precision; 5) included additional details in the figure legend and Methods section; 6) corrected all grammatical errors in the manuscript.

We believe that we have satisfactorily addressed all the concerns raised by the reviewers. For more details, please see the attachment.

We appreciate the time and effort you have put into this manuscript. We eagerly await your decision.

Sincerely,

Yu Wang and Bin Qu

Reviewer 2 Report

Please see the comments in the attachment. 

Author Response

Dear Reviewers,

We are submitting the revised version of our manuscript (ijms-2215600), titled ‘Bioactive Constituents of Verbena Officinalis Alleviate In-flammation and Enhance Killing Efficiency of Natural Killer Cells’ for consideration of publication in the International Journal of Molecular Sciences.

We are grateful to the reviewers for your valuable and constructive feedback. To address the reviewers' concerns, we have: 1) assessed the potential toxicity of the doses of VO-extract in mice (new Supplementary Fig. 1); 2) determined the relative contents of the five Verbena officinalis bioactive constituents (new Supplementary Fig. 2) and conductedadditional killing assays with a mixture of Verbenalin plus the three negative bioactive constituents (Acteoside, Apigenin, and Kaempferol) (new Supplementary Fig. 3); 3) added a prediction of the interaction between Verbenalin and NK receptor NKG2A and KIR2DL1 using an in silico molecular docking analysis (new Supplementary Fig. 4); 4) re-written the Discussion section to improve the cohesiveness, clarity, and precision; 5) included additional details in the figure legend and Methods section; 6) corrected all grammatical errors in the manuscript.

We believe that we have satisfactorily addressed all the concerns raised by the reviewers. For more details, please see the attachment.

We appreciate the time and effort you have put into this manuscript. We eagerly await your decision.

Sincerely,

Yu Wang and Bin Qu

Round 2

Reviewer 1 Report

I would like to accept the manuscript in its present format since the authors have addressed all my concerns satisfactorily. 

Author Response

Dear Reviewer #1,

We would like to express our sincere gratitude for your endorsement of the publication of our manuscript. We are truly grateful for the time and effort you, as well as the other reviewers, have dedicated to providing feedback and recommendations for improvement. Your insights and comments have been invaluable in enhancing the quality of our work.

Thank you again for your invaluable contribution to this work.

Sincerely,

Yu Wang and Bin Qu

Reviewer 2 Report

The authors have improved the manuscript. However, comment #8 was not addressed. The results at 10uM treatment of Verbenalin should be included in Figure 6. The writing of the manuscript is also awkward at times, with substantial grammatical and spelling errors (i.e. Lines 127, 207-208, 232, 238, 251, 381 etc.). Please send the paper for proofreading.  

Author Response

Dear Reviewer#2,

We sincerely appreciate the valuable and constructive feedback provided by you. In response to these comments, we conducted further live cell imaging experiments using NK cells treated with 10 µM of Verbenalin. We analyzed the same parameters as for the case of 30 µM, including the time required for killing, the number of killing events per NK cells, and the frequency of serial killers, single killers, and non-killers (see new Supplementary Fig. 4). Additionally, we have thoroughly reviewed the manuscript with assistance from a native English-speaking colleague and made all necessary corrections to grammatical and spelling errors. We believe that we have satisfactorily addressed all the concerns raised by you. For more details, please refer to the enclosed Response Letter.

We appreciate the time and effort you have put into this manuscript. We eagerly await your decision.

Sincerely,

Yu Wang and Bin Qu

Round 3

Reviewer 2 Report

Please check spelling errors throughout the manuscript.